# Improved Models of Imaging of Skylight Polarization Through a Fisheye Lens

**DOI:** 10.3390/s19224844

**Published:** 2019-11-07

**Authors:** Shaobo Sun, Jun Gao, Daqian Wang, Tian Yang, Xin Wang

**Affiliations:** School of Computer and Information, Hefei University of Technology, Hefei 230009, China; 2017110921@mail.hfut.edu.cn (S.S.); gaojun@hfut.edu.cn (J.G.); wangdq@mail.hfut.edu.cn (D.W.); 2017170722@mail.hfut.edu.cn (T.Y.)

**Keywords:** skylight polarization pattern, imaging theories, distortion, imaging models

## Abstract

Researchers have found that some animals can use the skylight polarization pattern for navigation. It is also expected to use the skylight polarization pattern for human navigating in the near future. However, the challenge is that the need for a more accurate and efficient model of the imaging of skylight polarization is always felt. In this paper, three improved models of imaging of skylight polarization are proposed. The proposed models utilize the analysis of the distribution of the skylight polarization pattern after the polarization imaging system. Given that the skylight polarization pattern after the polarization imaging system is distorted, the focus of this paper is on the degree of distortion of the skylight polarization pattern in these imaging models. Experiments in clear weather conditions demonstrate that the proposed model operates close to the actual acquired skylight polarization pattern.

## 1. Introduction

Once the skylight enters the atmosphere, it is polarized by the scattering and absorption of the atmosphere during the transmission process, and a polarization pattern of the skylight with stable distribution is formed [1]. The polarization information contained in this polarization pattern of the skylight can be used as a source of different animal navigation information. For example, desert ants [2], locusts [3], beetle [4], etc., can use the structure of their unique compound eye to sense the polarization pattern of the skylight and provide accurate compass information to achieve navigation capabilities. In addition, some underwater animals also navigate with polarization information. Unlike the skylight polarization pattern, the underwater polarization pattern is much weaker. The light from the sun and sky enters the water by refracting through the surface, and then the light scatters and refracts in the water, eventually forming the underwater polarization pattern [5,6,7]. Bionic polarized light navigation is an autonomous navigation method based on the visual perception of highly sensitive polarization. This method realizes the acquisition of navigation information by detecting and calculating the skylight polarization pattern [8,9].

Inspired by the structure of biological compound eyes, scholars have developed a variety of polarized light navigation sensors, which can be divided into two categories: ‘Point-source’ polarized navigation sensors and polarized imaging sensors. The first type is a non-imaging sensor, which realizes navigating and positioning by detecting the polarization information of the zenith apex [10,11]. The latter detects the polarized light by imaging and then analyzes the polarized image for obtaining navigation information [12]. Polarized vision sensors can acquire more polarization information from different directions of sky dispersion compared to the ‘Point-source’ polarized navigation sensors. However, in order to obtain full-sky polarization information, it is necessary to match the fisheye lens on the polarization imaging navigation sensor. Accordingly, Voss [13,14] designed a prototype to capture the skylight polarization pattern. On the other hand, in this research, the fisheye lens was used to achieve 180° celestial observations. Horvath [15,16] provided a full-sky imaging polarimeter that includes three fisheye lenses and three cameras. In this study, polarization modes were analyzed and measured using a 180° imaging polarization measurement. Pust and Shaw [17,18] used a 180° fisheye lens to image the sky onto a 1 million-pixel CCD camera through a polarized optical mirror. In addition, a full-sky imaging polarimeter was designed in this work, and then some experimental studies were carried out. The above work focuses on the practical observation of the skylight polarization pattern. On the other hand, the simulation and analysis of the theoretical polarization pattern can provide basic theoretical support for related applied research.

In previous studies, researchers have attempted to find a more precise description of the skylight polarization pattern and extract information that can be used for polarized light navigation. Therefore, the development of two different models has been promoted: The first type of model is mainly used to analyze the structure of the Earth’s atmosphere and dynamic information. In addition, it can solve the vector radiation transfer equation (VRTE) for the construction of the skylight polarization pattern. This modeling can cover the matrix operator theory [19], the discrete ordinate theory [20], the spherical harmonics theory [21], and the multiple scattering theory [22]. From a computational point of view, this type of model is extremely complex and depends on the initial conditions. Another type of model creates a simple and practical polarization pattern by analyzing the characteristics of the skylight distribution. For example, the single-scattering Rayleigh model, which shows the distribution characteristics of the polarization pattern in an ideal atmospheric environment [1], the singularity distribution characteristic model, which represents the skylight polarization pattern proposed by Berry [23], the analytical model of the polarization pattern under light intensity and good weather conditions by Wilkie [24], or the analytical model combined with Dennis and Berry’s singularity theory and Perez intensity developed by Wang Xin [25].

When the existing skylight polarization pattern is mapped from a three-dimensional space to a two-dimensional plane, the mapping method is completely different from the actual imaging principle. At the same time, in the actual experiment, the fisheye lens is installed in front of the acquisition system to get the skylight polarization pattern of the full-sky. Therefore, in order to extract the information that can be used for the navigation of polarized light more accurately in the skylight polarization pattern, the established sky polarization model should be combined with the imaging system to fully consider the distortion caused by the imaging system.

In this paper, the difference between the theoretical model and the actual detection and the causes of the difference from the imaging aspect has been analyzed based on the Rayleigh model. Three improved models based on imaging theories of skylight polarization have been proposed and the distortion of the models has been analyzed. On the other hand, the accuracy of one of the most common imaging modes (equidistance imaging mode) has been investigated. In this paper, the combination of the skylight polarization pattern and actual imaging has been achieved, which can match the detection results of different fisheye lenses. As has been analyzed, the experimental results show that the proposed model is closer to the actual measurement of the skylight polarization pattern compared to two-dimensional Rayleigh model.

## 2. Methods

### 2.1. Method for Obtaining the Skylight Polarization Pattern

The instrument used in this experiment was a custom-built system known as the full-sky skylight polarization pattern imaging system, as shown in Figure 1. The full-sky skylight polarization pattern imaging system was mainly composed of the optical path structure, imaging structure, and back-end system control software. The model of the fisheye lens was Sigma 8 mmF/3.5, and the effective field of view of the system was about 140°. The angle of intersection of the polarization axes of the three CCDs with the reference direction was set to 0, 60, and 120°. The geographic north direction was flush with the camera spindle.

It is worth noting, the Stokes vector method is the most common polarization detection algorithm. The polarization imaging model can be described by the Stokes vector of atmospheric light, where *I* is the radiation intensity, *Q* and *U* represent the linearly polarized light in two directions, and *V* represents the circularly polarized component. In addition, the degree of polarization (*Dop*) and the angle of polarization (*Aop*) can be expressed by Equation (1).
(1)$$\left\{ \begin{array}{l} Dop=\frac{\sqrt{Q^{2}+U^{2}+V^{2}}}{I}\\ Aop=\frac{1}{2}\tan^{-1}(\frac{U}{Q}) \end{array}\right.$$

Moreover, since the proportion of circular polarization in the atmospheric environment is small, V=0 is usually assumed in Equation (2). Therefore, *Dop* can be expressed as follows:(2)$$Dop=\frac{\sqrt{Q^{2}+U^{2}}}{I}$$

### 2.2. Rayleigh Skylight Polarization Pattern and Its Representation

Based on years of research on the skylight polarization pattern, the Rayleigh model can describe the distribution of the polarization pattern under ideal conditions [26]. It is assumed that the atmospheric polarization mode is mainly formed by single scattering of particles under clear weather conditions. The coordinate system should be established when establishing polarization mode, as shown in Figure 2.

In the coordinate system above, the celestial radius is 1, *S* stands for the solar position, *O* stands for the observer position, *Z* stands for the zenith, the *Z*-axis is toward the zenith, the *X*-axis is the east direction, and the *Y*-axis is the north direction. P(r,θ,φ) stands for any point on the sphere, θ stands for the zenith angle, and φ stands for the azimuth. The position of the solar space is expressed as $S(r,\theta_{s},\varphi_{s})$, where $\theta_{s}$ stands for the zenith angle of the sun, $\varphi_{s}$ stands for the azimuth angle of sun, and $h_{s}=90^{{}^{\circ}}-\theta_{s}$ stands for the elevation angle of the sun. *Aop*
α at any point in the celestial sphere is given by Equation (3).
(3)$$\tan\alpha =\frac{\sin\theta \cos\theta_{s}-\cos\theta \cos(\varphi -\varphi_{s})\sin(\theta_{s})}{\sin(\varphi -\varphi_{s})\sin(\theta_{s})}$$

The scattering angle $\gamma =\mathrm{acos}(\cos\theta \cos\theta_{s}+\sin\theta \sin\theta_{s}\cos(\varphi -\varphi_{s}))$, and *Dop* of the scattered light is Equation (4).
(4)$$P=P_{\max}\sin^{2}\gamma /\left(1+\cos^{2}\gamma \right)$$

In Equation (4), $P_{\max}$ represents the maximum *Dop* (the theoretical value is 1).

Figure 3 shows the distribution of *Aop* and *Dop* when the solar elevation and azimuth angles under the Rayleigh model are 0° and 90°, respectively.

### 2.3. Improved Models Based on Imaging Theories of Skylight Polarization

In a typical optical system, both the imaging and the actual model follow a similar theory [24]. This means that when the object is at a close distance, the imaging height and the actual height of the object follow Equation (5).
(5)$$y_{0}=\beta y$$

When the object is at infinity, the imaging height and the actual height of the object are given by Equation (6)
(6)$$y_{0}=f\tan\omega$$

In Equations (5) and (6), $y_{0}$ represents the ideal image height, β represents the lateral magnification, is a specified value, y shows the actual object height, f stands for the focal length of the fisheye lens, and f varies with the choice of the fisheye lens, and ω stands for the half angle of view of the lens. When the skylight polarization pattern is actually acquired by shooting the sky, the sky is approximated as “the object is at infinity”. Therefore, ideal similar imaging should follow Equation (6).

With respect to Equation (6), it is known that tanω→∞ and $y_{0}\to \infty$ when $\omega \to 90^{{}^{\circ}}$. Therefore, if the imaging is still performed according to the similar imaging theory in the imaging of the skylight polarization pattern, then the infinity of the imaging plane will appear when the field of view reaches 90°. Therefore, it is necessary to quote the “non-similar imaging” theory. The ideal size of the imaging plane is achieved by sacrificing the similarity of imaging.

When the skylight polarization pattern is actually collected, the non-similar imaging is realized through the distortion of the fisheye lens, thereby achieving the idealization of the imaging plane size. Therefore, in constructing the polarization pattern, it is necessary to analyze the distortion principle of the fisheye lens, and further construct the improved models based on imaging theories of skylight polarization. The imaging system of sunlight that passes through the fisheye lens is shown in Figure 4.

Different from normal lenses, the fisheye lens changes the mapping mode of the skylight polarization pattern. The light bends after passing the fisheye lens, which causes migration mapping. Through this migration mapping, we can finally obtain the image with a larger field of view. As shown in Equation (7), *r*, *r*_1_, *r*_2_, *r*_3_ and *r*_4_ in mapping functions are the corresponding distances of perspective imaging, stereographic imaging, equidistance imaging, equisolid angle imaging, and vertical projection between the image points and principal point.
(7)$$\begin{array}{ll} r=f\tan\theta & \;(\mathrm{perspective}\;\mathrm{imaging})\\ r_{1}=2f\tan\frac{\theta }{2} & \left(\mathrm{stereographic}\;\mathrm{imaging}\right)\\ r_{2}=f\theta & (\mathrm{equidistance}\;\mathrm{imaging})\;\\ r_{3}=2f\sin\frac{\theta }{2} & \left(\mathrm{equisolid}\;\mathrm{angle}\;\mathrm{imaging}\right)\\ r_{4}=f\sin\theta & \left(\mathrm{vertical}\;\mathrm{projection}\right) \end{array}$$

References [27,28,29] describe in detail the whole process of fisheye lens mapping three-dimensional information to two-dimensional imaging plane by changing the optical path. The schematic description of different mapping for fisheye lens are illustrated in Figure 5a, and the difference between a pinhole lens and a fisheye lens is shown in Figure 5b.

In actual fisheye lens imaging, the specific imaging modes can be divided into equidistance imaging mode, equisolid angle imaging mode, stereographic imaging mode, etc. [30]. This section introduces these three imaging modes and creates improved models based on them. Each of them are analyzed in turn.

#### 2.3.1. Improved Model of Skylight Polarization in Equidistance Imaging Mode

In the equidistance imaging mode: $y_{0}=f\omega$, according to the distribution of the skylight polarization pattern, combined with the fisheye lens somehow from the equidistance imaging mode, the spatial coordinate system is established as shown in Figure 6.

In Figure 6, T stands for any point in the celestial sphere, $T^{\prime }$ stands for the point projected onto the two-dimensional plane according to the equidistance imaging mode of the fisheye lens, $90^{{}^{\circ}}-\theta$ stands for the elevation angle, in which θ is equal to the half angle ω of view of the lens, φ stands for the azimuth angle, and R stands for the celestial radius.
(8)$$OT^{\prime }=y_{0}=f\theta$$

The spatial coordinates of any point T(x,y,z) in the celestial sphere can be calculated from the elevation angle and the azimuth angle, as in Equation (9), normalizing the radius of the celestial sphere and the imaging plane, specifically letting R=1, x and y multiply by the coefficient a. When θ=π/2, let ${\left(ax\right)}^{2}+{\left(ay\right)}^{2}=1$, solve for a=2/fπ, and Equation (9) for further Equation (10).
(9)$$\left\{ \begin{array}{l} x=Ry_{0}\cdot \cos\varphi \\ y=Ry_{0}\cdot \sin\varphi \\ z=R\cos\theta \end{array}\right.$$
(10)$$\left\{ \begin{array}{l} x=\frac{2}{\pi }\theta \cdot \cos\varphi \\ y=\frac{2}{\pi }\theta \cdot \sin\varphi \\ z=\cos\theta \end{array}\right.$$

Both sides of the equal sign of Equation (3) is multiplied by θ, and then any point T(x,y,z) of the celestial sphere can be imaged onto plane XOY and combined with Equation (10) to express the parameter φ with x and y, and further to obtain Equation (11).
(11)$$\tan\alpha \left(y\cos\varphi_{s}-x\sin\varphi_{s}\right)\sin\theta_{s}=\frac{2}{\pi }\sin\theta \cos\theta_{s}\cdot \theta -\cos\theta \left(x\cos\varphi_{s}+y\sin\varphi_{s}\right)\sin\theta_{s}$$

It can be solved by Equation (10), $\theta =\frac{\pi }{2}\sqrt{x^{2}+y^{2}}$, and it is brought to Equation (11), and the expression form of the angle of polarization α in the equidistance imaging mode can be obtained as Equation (12).
(12)$$\tan\alpha =\frac{\cos\left(\frac{\pi }{2}\sqrt{x^{2}+y^{2}}\right)\left(x\cos\varphi_{s}+y\sin\varphi_{s}\right)\sin\theta_{s}-\sin\left(\frac{\pi }{2}\sqrt{x^{2}+y^{2}}\right)\cos\theta_{s}\cdot \sqrt{x^{2}+y^{2}}}{\left(x\sin\varphi_{s}-y\cos\varphi_{s}\right)\sin\theta_{s}}$$

Similarly, the expression form of the scattering angle γ in the equidistance imaging mode can be obtained as Equation (13).
(13)$$\cos\gamma =\sin\theta_{s}\sin\left(\frac{\pi }{2}\sqrt{x^{2}+y^{2}}\right)\left(\cos\varphi_{s}\frac{x}{\sqrt{x^{2}+y^{2}}}+\sin\varphi_{s}\frac{y}{\sqrt{x^{2}+y^{2}}}\right)+\cos\theta_{s}\cos\left(\frac{\pi }{2}\sqrt{x^{2}+y^{2}}\right)$$

The degree of polarization is calculated as Equation (4).

According to Equation (12) and Equation (13), the improved model of skylight polarization is obtained based on the equidistance imaging mode, as shown in Figure 7a,b.

#### 2.3.2. Improved Model of Skylight Polarization in Equisolid Angle Imaging Mode

In the equisolid angle imaging mode: $y_{0}=2f\sin\frac{\theta }{2}$. In Figure 6, $OT^{\prime }=y_{0}=2f\sin\frac{\theta }{2}$, after normalizing the radius of the celestial sphere and the imaging plane, the spatial coordinates of any point T(x,y,z) on the celestial sphere are given by Equation (14).
(14)$$\left\{ \begin{array}{l} x=\sqrt{2}\sin\frac{\theta }{2}\cdot \cos\varphi \\ y=\sqrt{2}\sin\frac{\theta }{2}\cdot \sin\varphi \\ z=\cos\theta \end{array}\right.$$

The two sides of the equal sign of Equation (3) are multiplied by $\sin\frac{\theta }{2}$, and then any point T(x,y,z) of the celestial sphere can be imaged onto plane XOY and combined with Equation (13) to express the parameter φ with x and y, and further to obtain Equation (15).
(15)$$\tan\alpha \cdot \left(y\cos\varphi_{s}-x\sin\varphi_{s}\right)\sin\theta_{s}=\sqrt{2}\sin\theta \cos\theta_{s}\cdot \sin\frac{\theta }{2}-\cos\theta \left(x\cos\varphi_{s}+y\sin\varphi_{s}\right)\sin\theta_{s}$$

It can be solved by Equation (14), $\sin^{2}\frac{\theta }{2}=\frac{x^{2}+y^{2}}{2}$, $\cos\theta =1-\left(x^{2}+y^{2}\right)$, $\sin\theta =\sqrt{2\left(x^{2}+y^{2}\right)-{\left(x^{2}+y^{2}\right)}^{2}}$, and it is brought to Equation (15), and the expression form of the angle of polarization α in the equisolid angle imaging mode can be obtained as Equation (16).
(16)$$\tan\alpha =\frac{\left[1-\left(x^{2}+y^{2}\right)\right]\left(x\cos\varphi_{s}+y\sin\varphi_{s}\right)\sin\theta_{s}-\left(x^{2}+y^{2}\right)\cdot \sqrt{2-\left(x^{2}+y^{2}\right)}\cos\theta_{s}}{\left(x\sin\varphi_{s}-y\cos\varphi_{s}\right)\sin\theta_{s}}$$

Similarly, the expression form of the scattering angle γ in the equisolid angle imaging mode can be obtained as Equation (17).
(17)$$\cos\gamma =\sin\theta_{s}\sqrt{2-\left(x^{2}+y^{2}\right)}\left(x\cos\varphi_{s}+y\sin\varphi_{s}\right)+\cos\theta_{s}\left(1-\sqrt{x^{2}+y^{2}}\right)$$

The degree of polarization is calculated as Equation (4).

According to Equations (16) and (17), the two-dimensional improved model of skylight polarization is obtained based on the equisolid angle imaging mode, as shown in Figure 7c,d.

#### 2.3.3. Improved Model of Skylight Polarization in Stereographic Imaging Mode

In the stereographic imaging mode: $y_{0}=2f\tan\frac{\theta }{2}$. In Figure 6, $OT^{\prime }=y_{0}=2f\tan\frac{\theta }{2}$, after normalizing the radius of the celestial sphere and the imaging plane, the spatial coordinates of any point T(x,y,z) on the celestial sphere are given by Equation (18).
(18)$$\left\{ \begin{array}{l} x=\tan\frac{\theta }{2}\cdot \cos\varphi \\ y=\tan\frac{\theta }{2}\cdot \sin\varphi \\ z=\cos\theta \end{array}\right.$$

As can be seen, both sides of the equal sign of Equation (3) are multiplied by $\tan\frac{\theta }{2}$. Subsequently, any point T(x,y,z) of the celestial sphere onto plane XOY can be imaged. Then, it is combined with Equation (18). Furthermore, Equation (19) can be obtained.
(19)$$\tan\alpha \cdot \left(y\cos\varphi_{s}-x\sin\varphi_{s}\right)\sin\theta_{s}=\sin\theta \cos\theta_{s}\cdot \tan\frac{\theta }{2}-\cos\theta \left(x\cos\varphi_{s}+y\sin\varphi_{s}\right)\sin\theta_{s}$$

It can be solved by Equation (18), $\sin\theta \tan\frac{\theta }{2}=\frac{2\left(x^{2}+y^{2}\right)}{1+x^{2}+y^{2}}$, $\cos\theta =1-\frac{2\left(x^{2}+y^{2}\right)}{1+x^{2}+y^{2}}$, and it is brought into Equation (19). In addition, the expression form of the angle of polarization α in stereographic imaging mode can be obtained as in Equation (20).
(20)$$\tan\alpha =\frac{\left(1-x^{2}-y^{2}\right)\left(x\cos\varphi_{s}+y\sin\varphi_{s}\right)\sin\theta_{s}-2\left(x^{2}+y^{2}\right)\cos\theta_{s}}{\left(1+x^{2}+y^{2}\right)\left(x\sin\varphi_{s}-y\cos\varphi_{s}\right)\sin\theta_{s}}$$

Similarly, the expression form of the scattering angle γ in stereographic imaging mode can be obtained as in Equation (21).
(21)$$\cos\gamma =\sin\theta_{s}\frac{2}{1+x^{2}+y^{2}}\left(x\cos\varphi_{s}+y\sin\varphi_{s}\right)+\cos\theta_{s}\left(\frac{1-x^{2}-y^{2}}{1+x^{2}+y^{2}}\right)$$

The degree of polarization is calculated as in Equation (4).

According to Equations (20) and (21), the two-dimensional improved model of skylight polarization based on the stereographic imaging mode are obtained, as shown in Figure 7e,f. Besides, as shown in Figure 7, different operations have been made to compare the improved polarization model with the fisheye lens and without the fisheye lens.

### 2.4. Distortion Analysis of the Improved Models of Skylight Polarization Based on Imaging Theories

The above three two-dimensional improved models of the skylight polarization pattern correspond to the theory of three imaging systems, so that the three-dimensional representation of the skylight polarization pattern is reasonably described in the two-dimensional imaging system. However, the entry of sunlight into the fisheye lens leads to the incidence of distortion. Therefore, the skylight polarization pattern is also distorted after passing through the fisheye lens imaging system.

Measures of the magnitude of the distortion produced by the fisheye lens are expressed as radial magnification $\beta_{r}$ and tangential magnification $\beta_{t}$, as shown in Equation (22). Subsequently, after imaging by the fisheye lens in the three imaging modes, the degree of distortion of the skylight polarization pattern is quantitatively analyzed by referring to Figure 8.
(22)$$\left\{ \begin{array}{l} \beta_{r}=\frac{P^{\prime }Q^{\prime }}{PQ}\\ \beta_{t}=\frac{P^{\prime }S^{\prime }}{PS} \end{array}\right.s$$

The radial magnification $\beta_{r}$ in equidistance imaging mode is given by Equation (23)
(23)$$\beta_{r}=\frac{dy_{0}}{Rd\theta }=\frac{f}{R}=f$$

Moreover, the tangential magnification $\beta_{t}$ in equidistance imaging mode can be calculated by Equation (24)
(24)$$\beta_{t}=\frac{y_{0}d\varphi }{\sin\theta d\varphi }=\frac{y_{0}}{\sin\theta }=f\left(\frac{\theta }{\sin\theta }\right)$$

In the equidistance imaging mode, the radial magnification of the skylight polarization pattern is a fixed value of $\beta_{r}=f$. In addition, any point in the sky corresponds to the same radial magnification on the two-dimensional imaging plane. The tangential magnification $\beta_{t}$ is modulated by $\frac{\theta }{\sin\theta }$. Moreover, it is inversely proportional to θ. Hence, it can be said that when θ≈0, $\beta_{t}=f$, and when $\theta \approx \frac{\pi }{2}$, $\beta_{t}=1.57f$.

The radial magnification $\beta_{r}$ in equisolid angle imaging mode is given by Equation (25)
(25)$$\beta_{r}=\frac{dy_{0}}{Rd\theta }=f\cos\frac{\theta }{2}$$

In addition, the tangential magnification $\beta_{t}$ in equisolid angle imaging mode can be computed by Equation (26).
(26)$$\beta_{t}=\frac{2f\sin\frac{\theta }{2}}{R\sin\theta }=\frac{f}{\cos\frac{\theta }{2}}$$

In the equisolid angle imaging mode, $\beta_{r}$ and $\beta_{t}$ of the skylight polarization pattern are both modulated by $\cos\frac{\theta }{2}$, and the coefficients are reciprocal. Accordingly, while the former decreases with the increase of θ, the latter is directly proportional to θ. Therefore, when θ≈0, $\beta_{r}=1$ and $\beta_{t}=f$. In addition, when $\theta \approx \frac{\pi }{2}$, $\beta_{r}=0.707f$ and $\beta_{t}=1.414f$.

The radial magnification $\beta_{r}$ in stereographic imaging mode is given by Equation (27).
(27)$$\beta_{r}=\frac{f}{\cos^{2}\frac{\theta }{2}}$$

Additionally, the tangential magnification $\beta_{t}$ in stereographic imaging mode is obtained by Equation (28).
(28)$$\beta_{t}=\frac{2f\tan\frac{\theta }{2}}{R\sin\theta }=\frac{f}{\cos^{2}\frac{\theta }{2}}=\beta_{r}$$

In the stereographic imaging mode, $\beta_{r}$ and $\beta_{t}$ of the skylight polarization pattern are the same. In other words, both are modulated by $\cos^{2}\frac{\theta }{2}$, and directly proportional to θ. Consequently, when θ≈0, $\beta_{r}=\beta_{t}\approx f$, and when $\theta \approx \frac{\pi }{2}$, $\beta_{r}=\beta_{t}\approx 2f$.

The model of the skylight polarization pattern under three imaging systems has different distortion variables and is regulated by θ. This means that the larger θ the greater the distortion. As a result, the skylight polarization pattern is the smallest in the zenith region after the three imaging systems. In addition, the greatest distortion occurs near the horizon.

## 3. Results and Discussion

The improved models based on imaging theories of skylight polarization can effectively reduce the error caused by equipment in actual imaging. Additionally, it is able to reduce the distortion, which is closer to the actual measurement. Under the condition of clear weather, the atmospheric scattering model can be approximated as the Rayleigh scattering model. In this section, a comparison experiment is designed based on the Rayleigh scattering model to compare the actual measurement with the proposed model in this work and the two-dimensional representation model of the Rayleigh mode. Thereby, it is able to verify the validity of this model and the consistency of the spatial positional relationship between the model and the actual measurement.

### Experiment and Result Analysis

All the measurements in this paper were collected at the main teaching building of Hefei University of Technology. The latitude and longitude of this location are 31°50′49″N and 117°17′43″E, respectively. Furthermore, the experiment was performed from 8:30 to 17:00 on March 26, 2018, and the weather condition was fine. The imaging mode of the fisheye lens configured in the imaging system used in this study was equidistance imaging. Consequently, the two-dimensional simulation of the skylight polarization pattern of the proposed method is presented in the equidistance imaging mode. Moreover, in order to compare with the actual measurement, the effective field of view of the simulation was 140°. Additionally, the theoretical value of the sun position was obtained by the SPA (solar position algorithm) [31]. The *Aop* at each moment obtained by the actual measurement is shown in Figure 9a. The simulation of *Aop* is represented in Figure 9b. *AopEI* represents the simulation of the equidistance imaging method in the Rayleigh model, and *AopVP* represents the two-dimensional simulation of the vertical projection of the Rayleigh model. Likewise, the consistency of the simulation with the actual measurement is shown in Figure 9c.

In order to analyze the consistency between the theoretical model and the actual measurement considering time and space, the following treatment was performed: Under the same conditions, the stability of the *Aop* distribution is higher than that of the *Dop* [32,33,34]. Accordingly, in the present experiment, the *Aop* distribution was used to make the comparison. Using the simulated AopVP(i,j) and AopEI(i,j) pattern of the sky at a given sun position, the number of pixels can be counted that satisfies |AopVP( i , j) − Aop( i, j)|=∇AopVP and |AopEI( i , j) − Aop( i, j)|=∇AopEI. Then the number $N_{Rayleigh}$ of those celestial points is counted for which $\nabla AOP<AOP_{threshold}=5^{{}^{\circ}}$. In the present study, the proportion ε (similarity) of the sky that follows the Rayleigh model was introduced. This proportion is expressed as follows: $\varepsilon =N_{Rayleigh}/N$, where N=11335400 is the total number of pixels in the circular picture of the sky. The similarity between the two-dimensional simulation of the vertical projection of the Rayleigh model and the actual measurement is called as $\varepsilon_{AopVP}$. Likewise, the similarity between the simulation of the equidistance imaging method and the actual measurement is known as $\varepsilon_{AopEI}$. The comparison is made on the results followed by expressing through the degree of improvement ϑ, where $\vartheta =\varepsilon_{AopEI}-\varepsilon_{AopVP}$. It is worth mentioning that while comparing the actual measurement with the simulation of theoretical model, the addition and subtraction operations can be directly performed. Furthermore, the nonlinear errors that occur in the mapping relationship differences can be avoided. The results obtained from the comparison of six different times from 8:30 to 17:00 are represented in Table 1.

As can be clearly seen from the results in Table 1, the similarity $\varepsilon_{AopEI}$ is higher than the similarity $\varepsilon_{AopVP}$ at any time or at any solar elevation angle, and the improvement is about 20%. Although the experimental results have been greatly improved after using this model, the results of the simulation model somewhat deviate from the actual measurement. The main causes of this deviation are as follows:

(1) In the actual measurement, the increase in the solar elevation angle leads to a gradual increase in the ‘overexposed’ points caused by the sun, therefore these pixels cannot be reflected from the simulation model [34].

(2) The theoretical patterns of the angle of the polarization of skylight are calculated using the single scattering Rayleigh model. While, in the actual measurement, there is a large amount of Mie scattering in addition to single scattering. Therefore, there is a certain error between the actual measurement and the theoretical model [17,32].

(3) Although the improved model of skylight polarization in equidistance imaging mode is closer to the actual measurement, in addition to the central zenith region, the actual detection results still have errors caused by distortion. In addition, they gradually increase as the angle of view increases [27].

In the present study, in order to reduce the influence of the ‘overexposed’ points on the experimental results, caused by the sun, such a treatment was further performed: The anti-sun side was used as a comparison object. In order to reduce the error caused by the distortion of the actual measurement, the field of view was reduced to 100°. Additionally, the similarity and improvement were recalculated. The result is represented in Figure 10.

According to the above improvement, the average similarity between the proposed model in this paper and the actual measurement can reach 92.12%. Moreover, the average improvement can reach 30.33%. The results of similarity and improvement at six different times from 8:30 to 17:00 are shown in Table 2.

In addition, we also compared the two other imaging methods (equisolid angle imaging and stereographic imaging). The simulation results of *Aop* are presented in Figure 11a. *AopEA* represents the simulation of the equisolid angle imaging method in the Rayleigh model, and *AopS* represents the two-dimensional simulation of the stereographic imaging of the Rayleigh model. Likewise, the consistency of the simulation with the actual measurement is shown in Figure 11b.

The results of similarity and improvement at six different times from 8:30 to 17:00 are shown in Table 3. The similarity between the two-dimensional simulation of the equisolid angle imaging of the Rayleigh model and the actual measurement is called $\varepsilon_{AopEA}$. Likewise, the similarity between the simulation of the stereographic imaging method and the actual measurement is known as $\varepsilon_{AopS}$. And, $\vartheta_{EA}=\varepsilon_{AopEA}-\varepsilon_{AopVP}$, $\vartheta_{S}=\varepsilon_{AopS}-\varepsilon_{AopVP}$.

From the experimental results and Table 3, we can see that the other two models (skylight polarization in equisolid angle imaging mode and skylight polarization stereographic imaging mode) that have been improved by the fisheye lens are still better than the results without the fisheye lens.

We also made an improvement similar to Figure 10 for this experiment. The improved experimental results are shown in Figure 12.

The improved similarity and degree of improvement are shown in Table 4.

## 4. Conclusions

In this paper, the distribution of the skylight polarization pattern was investigated through the analysis of the imaging system of the skylight polarization pattern. In addition, the degree of distortion of the skylight polarization pattern was studied in the present work. Finally, improved models based on imaging theories of skylight polarization were established. Moreover, experiments show that the model was closer to the actual measurement, and the improvement of *Aop* was found to be as high as 20%. Eventually, after further processing the experimental results, the similarity could reach 92.12%, and the degree of improvement was over 30%. It is worth noting, the proposed model takes advantage of the fact that it is based on the actual imaging, which can more reasonably describe the distribution of atmospheric polarization modes in the real sky. Indeed, it can be helpful to achieve high navigation accuracy of polarization data calculation. It can provide richer and more accurate spatial distribution information for skylight polarization pattern. Furthermore, it is able to provide a new idea for establishing a more accurate analytical model of the skylight polarization pattern.

## Figures and Tables

**Figure 1 sensors-19-04844-f001:**
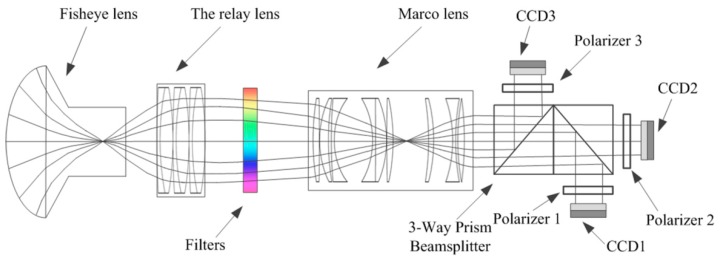
The full-sky skylight polarization pattern imaging system.

**Figure 2 sensors-19-04844-f002:**
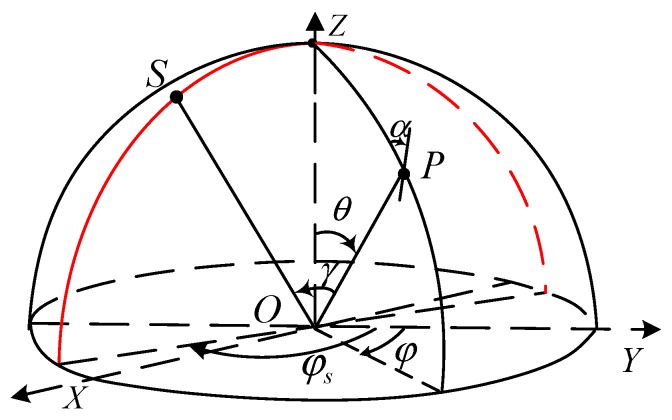
Geometric relationships of skylight polarization mode and beams.

**Figure 3 sensors-19-04844-f003:**
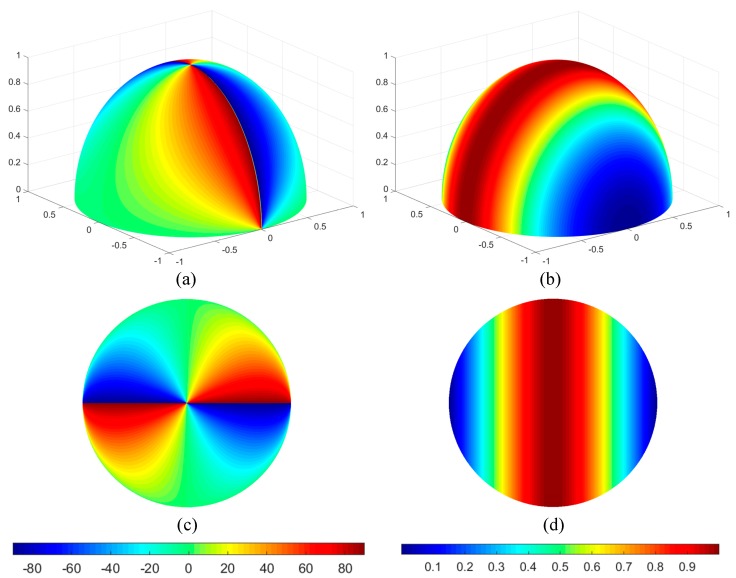
The Rayleigh model. (**a**) The *Aop* (angle of polarization) distribution of the Rayleigh model; (**b**) the *Dop* (degree of polarization) distribution of the Rayleigh model; (**c**) two-dimensional representation of the *Aop* of the Rayleigh model; (**d**) two-dimensional representation of the *Dop* of the Rayleigh model.

**Figure 4 sensors-19-04844-f004:**
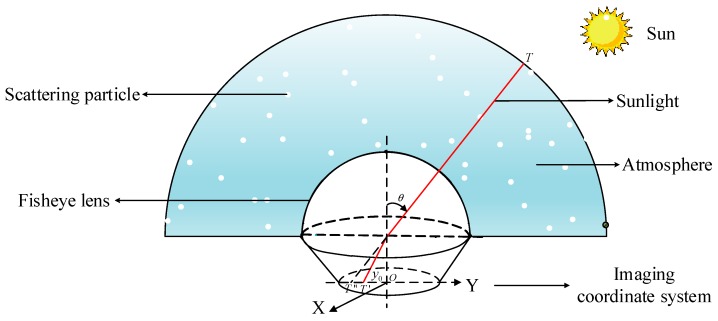
Schematic diagram of sunlight entering a polarization imaging system consisting of a fisheye lens and a CCD after scattering of particles in the atmosphere. T stands for any point in the celestial sphere, $T^{\prime }$ stands for the point projected onto the two-dimensional plane according to the equidistance imaging mode of the fisheye lens, $T^{\prime \prime }$ stands for the point of the ordinary lens, θ stands for the zenith angle of any point T of the celestial sphere, $y_{0}$ represents the distance from the point after equidistant imaging to the center of the imaging plane.

**Figure 5 sensors-19-04844-f005:**
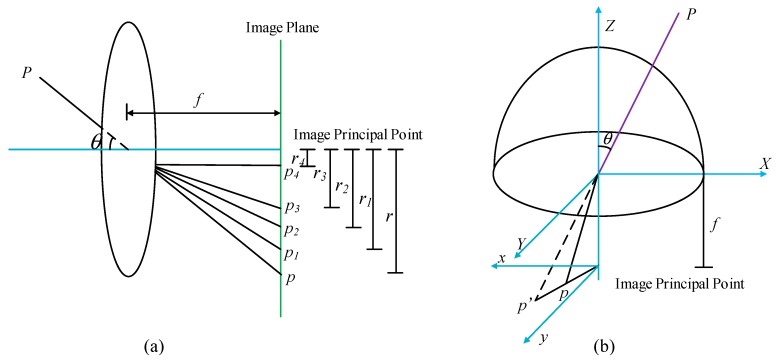
Mapping principles of different lenses. (**a**) The projections of different lens, p, p_1_, p_2_, p_3_ and p_4_ are perspective imaging, stereographic imaging, equidistance imaging, equisolid angle imaging, and vertical projection; (**b**) difference between a pinhole lens and a fisheye lens. For a fisheye lens, the actual image is the mapping of perspective image on hemisphere surface to image plane.

**Figure 6 sensors-19-04844-f006:**
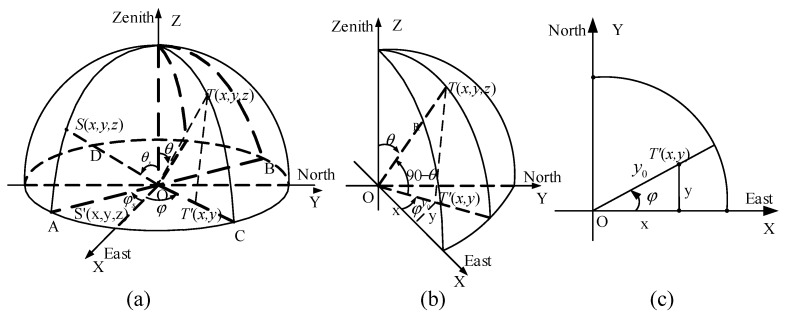
Theoretical skylight polarization pattern coordinate representation and imaging relationship. (**a**) The three-dimensional (3D) coordinate representation of skylight polarization pattern; (**b**) the meridian section of the celestial sphere; (**c**) the imaging relationship of skylight polarization pattern.

**Figure 7 sensors-19-04844-f007:**
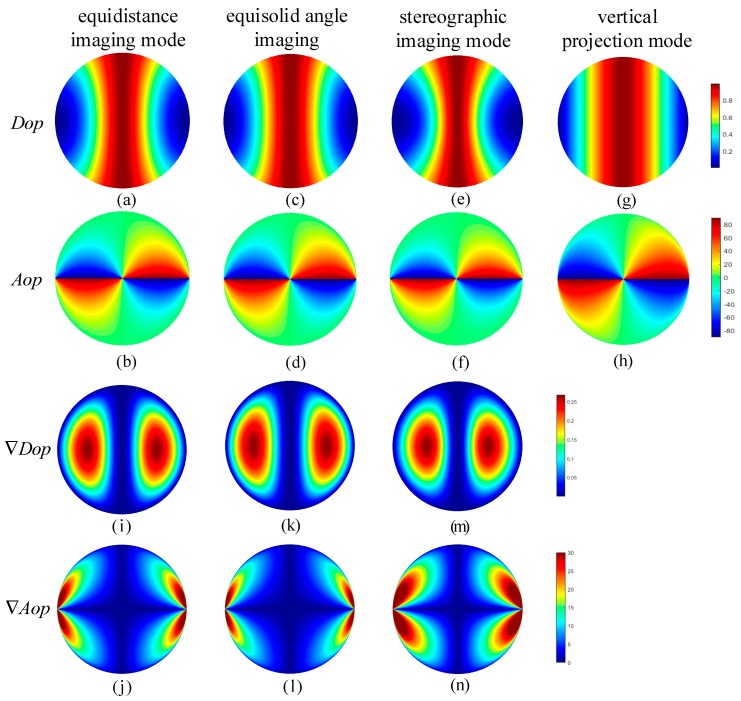
The improved models of skylight polarization in three imaging modes when the elevation angle is 0° and the azimuth angle is 90°; (**a**) *Dop* in equidistance imaging mode; (**b**) *Aop* in equidistance imaging mode; (**c**) *Dop* in equisolid angle imaging mode; (**d**) *Aop* in equisolid angle imaging mode; (**e**) *Dop* in stereographic imaging mode; (**f**) *Aop* in stereographic imaging mode; (**g**) *Dop* in vertical projection mode (without fisheye lens); (**h**) *Aop* in vertical projection mode (without fisheye lens); (**i**,**j**) the difference in skylight polarization pattern between the equidistance imaging mode and the vertical projection mode; (**k**,**l**) the difference in skylight polarization pattern between the equisolid angle imaging mode and the vertical projection mode;.(**m**,**n**) the difference in skylight polarization pattern between the stereographic imaging mode and the vertical projection mode.

**Figure 8 sensors-19-04844-f008:**
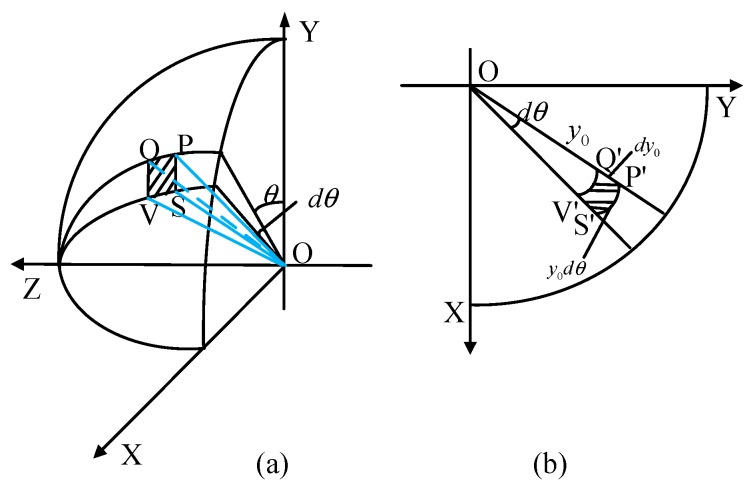
Correspondence between a tiny bin PQVS in the skylight polarization pattern and $P^{\prime }Q^{\prime }V^{\prime }S^{\prime }$ on the imaging plane, where light is incident from the *Z*-axis and imaged on the plane XOY after passing through the imaging system. (**a**) The three-dimensional coordinate representation of skylight polarization pattern; (**b**) the imaging relationship of skylight polarization pattern.

**Figure 9 sensors-19-04844-f009:**
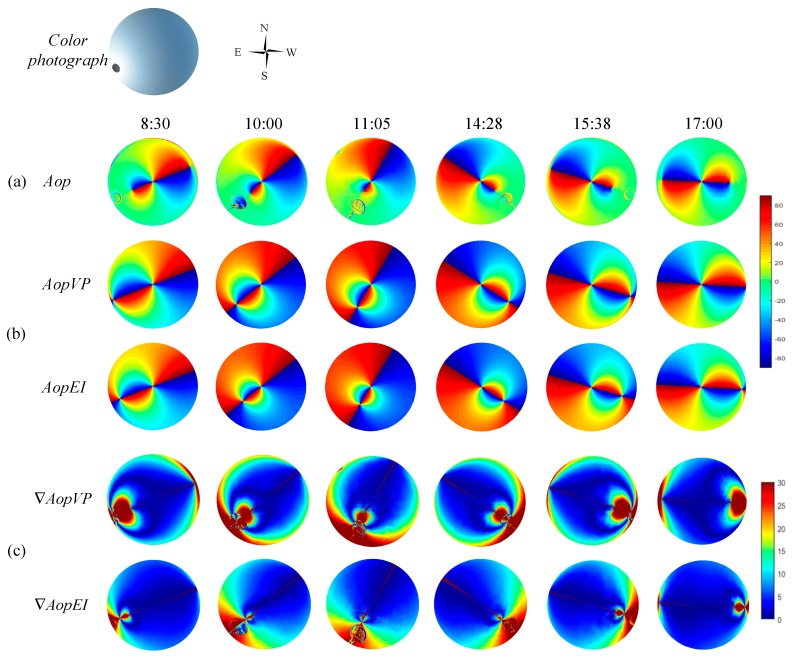
*Aop* distribution and its consistency representation of the actual measurement, simulation in equidistance imaging mode, and two-dimensional Rayleigh simulation; (**a**) the *Aop* at each moment obtained by the actual measurement; (**b**) the simulation of *AopVP* and *AopEI*, where *AopVP* represents the two-dimensional simulation of the vertical projection of the Rayleigh model, and *AopEI* represents the simulation of the equidistance imaging method in the Rayleigh model; (**c**) the consistency of the simulation with the actual measurement; where, ∇AopVP=|AopVP( i , j) − Aop( i, j)| and ∇AopEI=|AopEI( i , j) − Aop( i, j)|.

**Figure 10 sensors-19-04844-f010:**
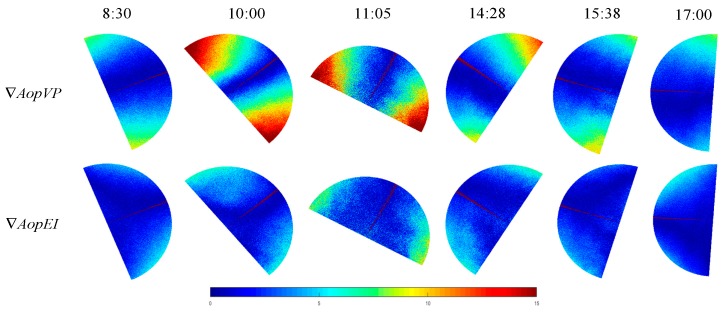
The result of consistency after experimental improvement.

**Figure 11 sensors-19-04844-f011:**
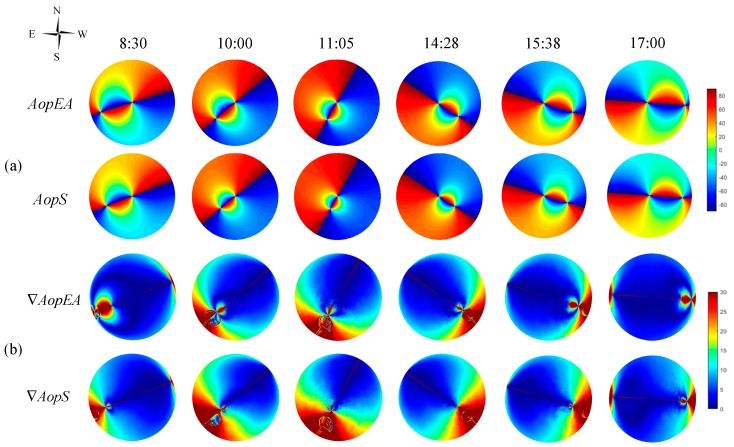
The consistency representation of the actual measurement, simulation in equisolid angle imaging and stereographic imaging mode; (**a**) the simulation of *Aop**EA* and *Aop**S*, where *Aop**EA* represents the two-dimensional simulation of the equisolid angle imaging of the Rayleigh model, and *Aop**S* represents the simulation of the stereographic imaging method in the Rayleigh model; (**b**) the consistency of the simulation with the actual measurement; where, ∇AopEA=|AopEA( i , j) − Aop( i, j)| and ∇AopS=|AopS( i , j) − Aop( i, j)|.

**Figure 12 sensors-19-04844-f012:**
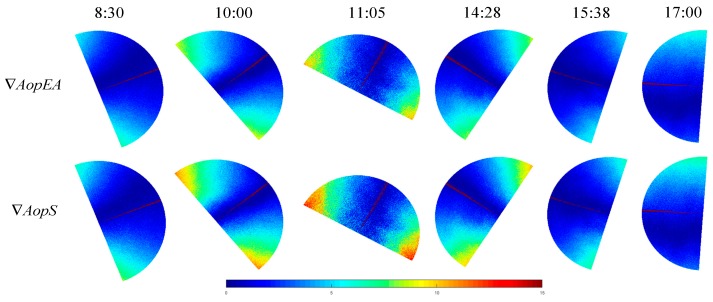
The result of consistency after experimental improvement.

**Table 1 sensors-19-04844-t001:** Similarity and improvement of equidistance imaging, two-dimensional Rayleigh simulation and actual measurement.

Time	$90^{{}^{\circ}}-\theta_{s}$	$\varepsilon_{AopEI}(\%)$	$\varepsilon_{AopVP}(\%)$	ϑ(%)
8:30	29.10	61.62	43.46	18.16
10:00	46.24	50.05	30.88	19.17
11:05	55.86	41.22	26.97	14.25
14:28	47.17	51.16	28.11	23.05
15:38	34.21	61.66	40.80	20.86
17:00	17.43	70.06	53.57	16.49

**Table 2 sensors-19-04844-t002:** Similarity and degree of improvement after experimental improvement.

Time	$90^{{}^{\circ}}-\theta_{s}$	$\varepsilon_{AopEI}(\%)$	$\varepsilon_{AopVP}(\%)$	ϑ(%)
8:30	29.10	97.90	73.78	24.12
10:00	46.24	89.17	42.77	46.40
11:05	55.86	80.08	48.99	31.09
14:28	47.17	90.06	53.71	36.35
15:38	34.21	98.65	71.23	27.42
17:00	17.43	96.86	80.25	16.61

**Table 3 sensors-19-04844-t003:** Similarity and improvement of equisolid angle imaging, stereographic imaging, two-dimensional Rayleigh simulation, and actual measurement.

Time	$90^{{}^{\circ}}-\theta_{s}$	$\varepsilon_{AopEA}(\%)$	$\varepsilon_{AopS}(\%)$	$\varepsilon_{AopVP}(\%)$	$\vartheta_{EA}(\%)$	$\vartheta_{S}(\%)$
8:30	29.10	55.38	50.42	43.46	11.92	6.96
10:00	46.24	44.53	33.25	30.88	13.65	2.37
11:05	55.86	35.17	28.13	26.97	8.20	1.16
14:28	47.17	44.26	36.46	28.11	16.15	8.35
15:38	34.21	58.36	43.72	40.80	17.56	2.92
17:00	17.43	68.59	59.28	53.57	15.02	5.71

**Table 4 sensors-19-04844-t004:** Similarity and degree of improvement after experimental improvement.

Time	$90^{{}^{\circ}}-\theta_{s}$	$\varepsilon_{AopEA}(\%)$	$\varepsilon_{AopS}(\%)$	$\varepsilon_{AopVP}(\%)$	$\vartheta_{EA}(\%)$	$\vartheta_{S}(\%)$
8:30	29.10	92.34	86.57	73.78	18.56	12.79
10:00	46.24	80.25	66.48	42.77	37.48	23.71
11:05	55.86	70.13	58.24	48.99	21.14	9.25
14:28	47.17	82.56	70.16	53.71	28.85	16.45
15:38	34.21	95.53	89.17	71.23	24.30	17.94
17:00	17.43	90.37	88.45	80.25	10.12	8.20

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
