# Peer review of "Improved Models of Imaging of Skylight Polarization Through a Fisheye Lens"

_sensors, 2019, doi:10.3390/s19224844_

Round 1
Reviewer 1 Report
The authors presented a very interesting and novel paper on modeling sky polarization patterns. The authors used Rayleigh scattering to model sky polarization phenomena and added new physics in their model to improve the performance of their algorithm. Their results are compared to measurements from fish eye lenses. It is a very important contribution to the literature and add to the sky/celestial polarization patterns.
There few minor questions:
Fish eye lens are notorious for changing the polarization patterns due to the internal optics. Can you discuss your observations in how your fish eye lens change polarization patterns? Please provide some data. There are also new evidence that polarization patterns are also present underwater. It would be good to cist some of this work and add it in your introduction. Discuss how is it similar and different to celestial polarization patterns.Author Response
Please see the attachment.

Reviewer 2 Report
This paper presents a mapping model from 3-D space to 2-D image when doing skylight polarization mode simulation, which is meaningful for the skylight polarization detection.
Based on the basic fish-eye lens models, the distortion caused by fish-eye lens during the imaging process is modeled, and the errors of different models are analyzed. Then the experiments are carried out to verify some of the proposed models.
The reviewer thinks the paper could be improved in the following aspects:
1. The terms in the paper should keep identical, for example, but not limited to “full-sky” and “all-sky”.
2. There are some spelling errors, for example,the published year of Ref. 22 should be 2016,not 2004.
3. The experiments seem inadequate. The experiments can only verify the mapping error of Aop image from 3-D space to 2-D image is significantly better than that of direct projection after using the fish-eye lens imaging model. There is no correction of the basic fish-eye lens model or the skylight polarization model for skylight polarization detection. In addition, there are no experiments involving the other two proposed models at all.
Round 2
Reviewer 2 Report
In this revised version, the experiments of the other two proposed models (equisolid angle imaging and stereographic imaging) are added. And the results show the improvement of the model.